# Developing an Automated System to Control the Rolled Product Section for a Wire Rod Mill with Multi-Roll Passes

Andrey A. Radionov [1], Olga I. Petukhova [2], Ivan N. Erdakov [3,*], Alexander S. Karandaev [4],
Boris M. Loginov [5] and Vadim R. Khramshin [2]

1   Department of Automation and Control, Moscow Polytechnic University, 38, Bolshaya Semyonovskaya Street, 107023 Moscow, Russia
2   Power Engineering and Automated Systems Institute, Nosov Magnitogorsk State Technical University, 38, Lenin Avenue, 455000 Magnitogorsk, Russia
3   Department of Metal Forming, South Ural State University, 454080 Chelyabinsk, Russia
4   Department of Information-Measuring Equipment, South Ural State University, 454080 Chelyabinsk, Russia
5   Department of Mechatronics and Automation, South Ural State University, 76, Lenin Avenue, 454080 Chelyabinsk, Russia
*   Correspondence: erdakovin@susu.ru; Tel.: +7-908-826-8619

**Abstract:** The key priority of metallurgical industry development is expanding the range and improving the quality of bar products and their major component, steel wire. Continuous wire rod mills with multi-roll passes have been developed and implemented over the past decades. These include mills with four-roll passes with mutually perpendicular rolls. The specific feature of mills with complex passes is the impossibility of conduct and the direct measurement of the workpiece dimensions in several directions during rolling. The paper studies the development of a system for indirect control over complex section geometry by adjusting the interstand space tension with simultaneous compensation for changes in rolling forces. The paper contribution is the first justification of a technique for the control over the indirect rolled product section on mills with multi-roll passes based on theoretical and experimental research. Analytical and experimental dependencies between the metal pressure on the rolls and the semifinished rolled product temperature, rolling speed, and single drawing have been obtained for various steel grades. The impact of process factors on the rolled product section geometry when rolling in stands with four-roll passes has first been studied. The automated control system implementing the proposed technique has passed pilot tests on a continuous five-stand mill. The processes occurring in closed-loop speed and torque control systems under controlling and disturbing effects have been experimentally studied. Implementing the proposed algorithms indirectly confirmed the reduced impact of tension and pressure on the section geometry.

**Keywords:** multi-roll passes; system for indirect control; rolling forces; geometry variation; algorithms; analytical and experimental dependencies

## 1. Introduction

Currently, two key techniques are used to produce metal wire: drawing through monolithic dyes [1], drawing through roller dyes [2] and rolling in multi-roll passes. The replacement of drawing with rolling allows lifting the single draft limitations, significantly improving the performance [3–5]. The application of passes is defined by favorable straining conditions and, as a result, improved mechanical properties of the wire [6–9]. Compared to drawing, drafting in passes is more energy-efficient. Therefore, developing techniques for rolling mills with multi-roll passes and their commercial implementation are urgent and promising problems.

Figure 1 shows a process flowsheet of a continuous *n*-stand rolling mill [10]. By the composition and equipment pattern, wire rod mills do not differ fundamentally from

tandem cold rolling mills [11]. Therefore, Figure 1 conditionally shows two-roll stands. As a rule, wire rod mills comprise an unwinder, a continuous group of rolling stands, and a winder.

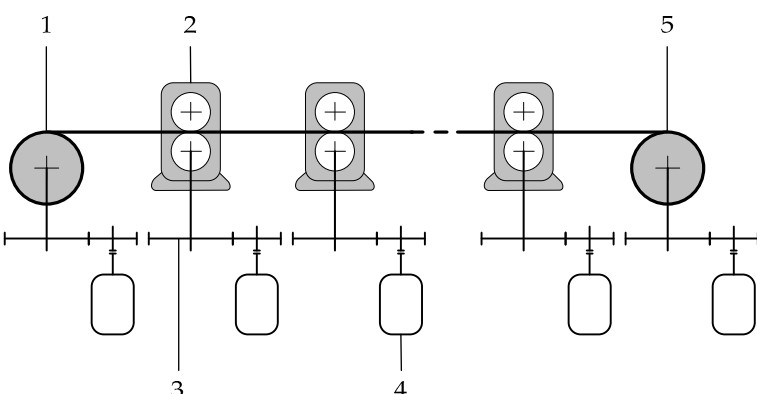

**Figure 1.** Process Flowsheet & Kinematic Diagram of The Continuous *n*-Stand Rolling Mill Drives: 1: unwinder; 2: rolling stand; 3: reducer; 4: motor; 5: winder.

The analysis of regulatory documents revealed the lack of any special requirements to the quality of wire produced on mills with multi-roll passes. In most cases, the requirements for wire rods are accepted as a standard. This assumption is justified since, according to the interstate standard GOST 2590-2006, a steel wire rod is a circular cross-section profile with a diameter of 5–9 mm, produced on wire rod mills and winded. This confirms the fundamental similarity of wire rods with a wire produced by rolling.

Improving the quality of wire rods and cold-rolled wire obtained from them is a global problem [12,13]. According to [14], innovative solutions will be developed in the 21st century to meet the requirements for the effective production of wire with excellent properties. Herewith, continuous rolling will involve more steel grades. The most important bar quality parameter is their cross-section accuracy. The requirements to the dimensional tolerances along the length are defined by standards and industry documents. A detailed analysis of international standards for the production of wire rods and wire on rolling mills is provided in [15]. Thus, according to the US standard A510, the maximum diameter deviation is ±0.4 mm for the entire range of sections within 5.5–19 mm. Threat, the ovality should not exceed 0.635 mm [16]. The Japanese standard JIS G3505:2004 (low carbon steel wire rod) specifies a rated diameter tolerance of 6–9% [17]. The ovality should not exceed half the sum of the tolerances. Note that the common trend is expanding the parameter variation ranges, which provides manufacturers with ample opportunities. Leading wire rod manufacturers claim the possibility of manufacturing them with tolerances of less than ±0.1 mm. Thus, Danieli has developed a high-speed technology for manufacturing wire rods with a diameter of 4.5 to 25 mm in coils weighing up to 3.5 tons with tolerances of ±0.075 mm and ovality of up to 50% tolerance [18].

According to [19], despite the introduction of new rolling processes and wire rod mill equipment, the issues of high rolled product accuracy cannot be considered resolved to the full extent. The problem of achieving stable geometry along the rolled product length requires further study. The greatest difficulty in its solving is the need for strictly controlled interstand space tensions affecting the geometry. Therefore, the problem of providing the target section accuracy by purposefully improving the tension-based rolling modes is relevant.

The practice of operating continuous bar and wire rod mills stipulates for the empirical definition of optimal tensions which depends entirely on the staff qualifications and experience. Many designers prefer to eliminate or reduce tension whenever possible. Threat, automated drive speed control according to the minimum tension principle is applied. However, in some cases, longitudinal forces are unavoidable or desirable in continuous bar

rolling. As indicated below, the tension factor is used, e.g., to control the section along the rolled product length.

Stable bar geometry, including wire rods, is formed in all stands and affects the finished section. Therefore, the problem of stabilizing the section geometry in each stand is relevant for most wire rod mills. In the latest-generation mills, this problem is partially resolved by rolling blocks of various designs. Thereat, fluctuations in the semifinished rolled product dimensions in the continuous stand groups are most often eliminated by tension adjustment. Thus, according to the empirical data-based conclusion in [19], interstand tension is an important process parameter determining rolling stability and section accuracy.

The existing techniques for considering tensions and their impacts in section rolling are classified as follows [19]:

- Techniques to compensate for the impact of tensions by changing the strain parameters. They allow for obtaining a shape change compensating for the longitudinal force impact when varying the workpiece parameters or straining modes under the particular mill conditions.
- Techniques to minimize tensions. They are used in special systems which facilitate the actual elimination of the tension impact. Tension is minimized (within certain limits) using the algorithms for automated control of the stand group speed mode.
- Indirect techniques to provide rational tensions by selecting the drive speed ratios. They are used to stabilize the section geometry in continuous stand groups.

The listed techniques have a positive effect under certain predetermined rolling conditions. However, it is extremely difficult to provide such conditions (no fluctuations in the workpiece geometry, the specified drive speed limits, and constant tension). Therefore, the problem of flexibly and effectively controlled section accuracy in the tension control mode remains relevant. This will allow the building of efficient automated geometry control systems, regardless of the rolled product section.

The basic principles of indirect tension control are based on the research findings and applications published in the middle-end of the last century [20–22]. These comprise the ways of automated control over free rolling currents and minimum (zero) tension. Refs. [23–25] have theoretically and experimentally proven that these techniques provide (with a certain accuracy) constant geometry and temperature profile along the entire workpiece length.

The systems based on the free-rolling current control are also used to limit the force interaction between the stands (without loopers) of section and broad-strip hot rolling mills [26–30]. They limit excessive tension or backpressure (negative tension) by automatically maintaining the tension close to zero. As an example, [31] provides a quantitative estimation of the force interaction for the three-stand mill drives with fluctuations in rolling parameters. Furthermore, [32] developed a control system based on the improved load identification algorithm due to dynamic torque adaptation.

The analysis allows for reasonably stating that the development of an automated section control system for wires produced on continuous rolling mills is relevant. The control principle should be based on the indirectly adjusted interstand space tension. It will be shown below that in this case, systematic deviations in the metal pressure on the rolls should be compensated, arising under the action of uncontrolled process factors.

## 2. Problem Statement

### 2.1. Characteristics of Mills with Multi-Roll Passes

Rolling mills with multi-roll passes are classified by the number of the stand rolls forming the strain zone geometry, the pass type, and the multiplicity and kinematic diagram of torque transmission to the rolls. Mill designs with two-, three-, and four-roll passes are known. Refs. [33–35] characterize strains in three- and four-roll stands. Refs. [36,37] perform a comprehensive study of three multi-roll wire rod mill types. Laboratory experiments have been performed to study the strain and load parameters of two-, three-, and four-

roll stands with the same roll diameter. As a result, four-roll mill superiority areas have been identified.

Various stand designs with two or all driven rolls are known. Figure 2a shows a picture of FUHR's four-roll mill designed for cost-effective wire production [38]. According to the figure, the axes of horizontal and vertical rolls are in the same plane but perpendicular to each other. Mills with four-roll passes have been commercialized at some Russian metallurgical plants. Thus, in [39], a unit to manufacture 6 mm diameter steel rods from a cast billet with a cross-section of 38 × 47 mm is described. Ref. [40] characterizes the aluminum and copper rod production unit. The history of the development of the considered class of mills is described in more detail in [41].

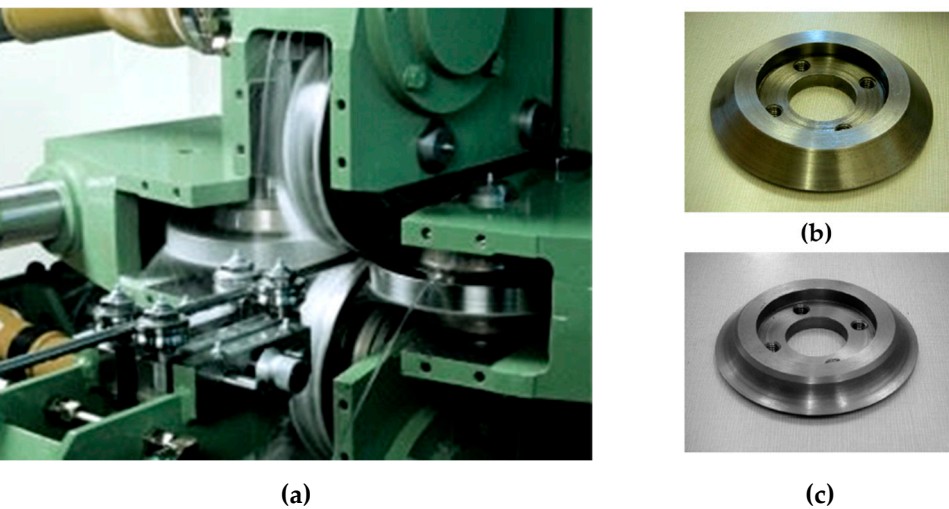

**(a)**  **(b)**  **(c)**

**Figure 2.** Rolling of Wire in a Four-Roll Pass Stand (**a**) and Rolls for Square (**b**) and Round (**c**) Sections.

Ref. [5] characterizes a five-stand wire rod mill with four-roll passes, operated at the Beloretsk Metallurgical Plant (Russia). The process flowsheet and the equipment are described in [3,42]. The mill was manufactured by SKET (Germany) and had been operating for a long time. It is designed for rolling 2.5–3.5 mm diameter pig wire from initial billets from –6.5–8.0 mm diameter wire rods. All further considerations in the paper are performed for this mill, so that the metallurgical plant is not further referred to. The increased interest in this unit is determined by its current reconstruction. This provides an opportunity to improve the technology and implement advanced control systems.

Table 1 provides four-roll pass options rolled on the mill [3]. When describing the technology, as a rule, intermediate passes are not mentioned, so the scheme was titled "Circle-Circle" for five-stand rolling. Draft by the "Circle-Square" and "Square-Circle" schemes is implemented by four-roll passes formed by conical crowns (Figure 2b) or radial grooves (Figure 2c) cut on rolls [43].

**Table 1.** Designations and Schemes of Four-Roll Passes.

| Pass | Circle-Square | Circle-Octagon | Octagon-Square | Octagon-Octagon | Octagon-Circle |
|---|---|---|---|---|---|
| Pass shape | ◇ | ⬡ | ⬡ | ⬡ | ⬡ |
| Drawing ratio | 1.57 | 1.11 | 1.66 | 1.17 | 1.05 |

According to the definition, drawing ratio (drawing) means the ratio of the workpiece lengths or cross-sections before and after rolling in the stand:

$$\mu = \frac{l_1}{l_0} = \frac{Q_0}{Q_1}, \tag{1}$$

where $l_0$, $Q_0$ are the workpiece length and cross-section before rolling (at the stand inlet); $l_1$, $Q_1$ are the workpiece length and cross-section after rolling (at the stand outlet).

Workpiece herein means an intermediate product between the primary billet (semi-finished rolling product) and the finished wire.

The reconstruction is aimed at improving the wire quality. This indicator is determined, first of all, by the workpiece section accuracy along the length. Hereinafter, the term workpiece indicates a semifinished product between the initial billet (semifinished rolled product) and the finished rolled product–wire. Thereat, the section shape changes after each stand (see Table 1). Irrespective of this, the section shape—a circle of a given diameter, inscribed in the profile, its minimum ovality, and the diameter constancy along the length should be provided. One of the reconstruction tasks is developing an indirect control system for the rolled product section to replace the outdated automated tension control system (ATCS).

In operating mills, electric drives, as a rule, are made according to the DC motor–thyristor converter principle [44]. Despite the wide use of AC motors, such drives are installed on many wire rod mills, including the one under study. However, reconstruction stipulates for replacing DC motors with asynchronous motors powered by frequency converters (FC-AM system).

During the mill operation, a unique experimental database has been built, and the rolling of various steel grades in a single or several stands has been studied. Complex couplings of electromechanical systems through metal have been studied. Some study results have been published earlier. Thus, in [3,10], the energy-power rolling parameters have been calculated based on the model of a strain zone formed by four rolls. Refs. [45,46] are devoted to improving automated drives and control systems. Ref. [4] dwells upon the development of digital models for interconnected electrical complexes of a wire rod mill. In [47–49], automated tension control systems have been developed for winders and unwinders. This paper considers the development of an automated control system for the rolled product section (RPSACS).

### 2.2. Analysis of Automated Tension Control Systems

The key metal forming problem is obtaining rolled products of the required shape with the target section geometry along the length. The section accuracy depends on many factors related to the mechanical equipment and the process [50]. The complexity of shaping a workpiece section at the intermediate stand output is mainly caused by the following:

1. The section geometry should be adjusted in several diagonal directions. In this case, the section shape changes after each stand (rectangle-octagon-square, etc.).

2. Installing meters capable of measuring the workpiece geometry in several directions in the mill process line is impossible. There are currently no such meters designed for industrial use. The exception is laser meters that may hardly be operated in the online rolling mode [51,52].

Therefore, the systems for the indirect control over the rolled product geometry are used at operating mills. Most known systems are built on the principle of automated control on either interstand tension or thickness and tension [31,53–55]. However, they involve directly measuring the metal pressure on the rolls. For stands with multi-roll passes, this approach is inapplicable. Refs. [32,56] study ATCS with the tension trackers. Ref. [57] proposes a technique to control tension with a tracker, considering the dynamic torque change. A tension control system has been developed for the continuous strip processing line, directly controlling the motor current. This is to provide a fast dynamic response in

the drive linear acceleration or deceleration modes. However, there is no practice of using tension trackers at wire rod mills.

The five-stand mill operation practice has identified that any change in rolling conditions in one or more stands affects the interstand tension. The most significant factors are geometry variation and temperature gradient of semifinished rolled product and change in the metal-roll contact friction conditions and the metal yield strength. In turn, tension fluctuations affect the workpiece geometry and section. Ref. [58] analyzes the impact of stand strain and interstand tension on the wire rod and bar section defects. The impact of tension on the rolled product geometry is also confirmed by the authors of [31,59].

For continuous mills, the loss of tension and interstand looping are risky modes. In this case, stability loss (pass stalling), breakage of guide unit and rolls, and other situations leading to accidents are possible. In [60], interstand tensions are calculated with a change in the section mill rolling process conditions. The process conditions excluding the loss or unacceptable increase in tension are determined.

The closest analogs to the ATCS of mills with multi-roll passes are the systems developed for drawing units. Their operation is underlain by the principle of maintaining zero tension by control over the stand motor torque [48,61]. For DC motors, these are zero current control systems [62]. Ref. [63] considers an automated control system for drive counter tension at direct-flow drawing mills based on the master and slave units. This system was adopted as a prototype in the development of RPSACS for the five-stand wire rod mill being considered.

The key drawback of the ATCS being studied is the failure to consider the issues of compensation for the tension impact on the metal pressure on the rolls during their development (for drawing units, such a problem cannot be set in principle). In addition, the authors failed to study the pressure impact on the rolled product geometry, which is important when rolling complex-section products. For these reasons, known systems will not provide the desired section control for wire rod mills with multi-roll passes.

The listed drawbacks of the known current and zero tension automated control systems predetermined the development of a new rolled product tension and section automated control system. The paper provides the results of developing such a system.

### 2.3. Characteristics of the Operating Five-Stand Mill ATCS

When the five-stand mill was put into operation, an indirect system for interstand tension control was installed. Its simplified circuit diagram is shown in Figure 3 [41]. It provides for stand division into the master and slave ones (for clarity, two of the four slave stands are shown). The master stand drive provides the target mill speed and compensates for inaccuracies in the current settings of a slave stand. The slave stand drives are built on the principle of indirect tension control [48]. In a simplified form, the tension control problem is reduced to the stabilization of motor currents.

The ATCS operating principle is based on regulating a current close to zero. A wire rod is loaded into the mill manually while the speed is controlled in all stands. After loading, the slave-stand motor currents are fixed. Then the mill is stopped, the workpiece dimensions are measured, and, where appropriate, the interstand tensions are corrected. Then, the mill is switched to the automatic mode while the current setting signals are fed to the slave-stand control systems. The speed controllers of these stand drives reach saturation. The current and speed controllers are set to the modular optimum [64].

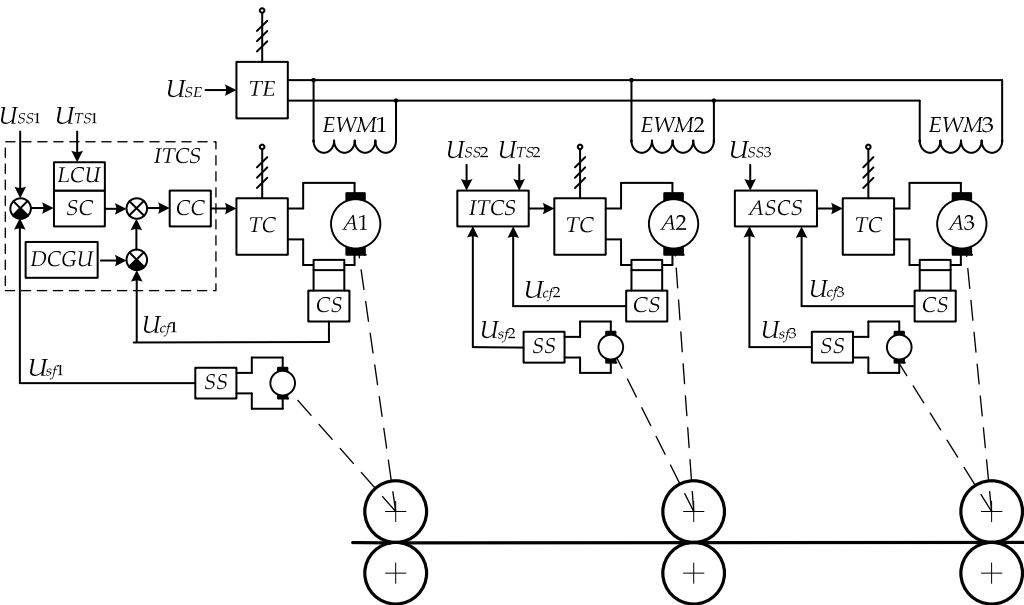

**Figure 3.** Functional Circuit of the Indirect Tension Control System: A1-A3, EWM1-EWM3:armatures and excitation windings of DC motors; TC: thyristor converter; TE: thyristor exciter; CC, SC: current and speed controllers; CS, SS: current and speed sensors; LCU: limit control unit; DCGU: dynamic current generation unit; ITCS: indirect tension control system; ASCS: automated speed control system; $U_{SSi}$, $U_{TSi}$: speed, tension, and excitation current setting voltages; $U_{cfi}$, $U_{sfi}$: current and speed feedback voltages.

The mill operation allowed for identifying the following drawbacks:

1. Low accuracy of the workpiece geometry control due to the lack of correction when the rolling conditions change.

2. Load change disturbances throughout the entire mill from the first to the last stand, associated with the unstable workpiece geometry and physical and mechanical properties and fluctuations in the wire rod tension at the unwinder output.

3. Lack of compensation for the impact of process disturbances on the metal pressure on the rolls. The metal pressure impact on the workpiece geometry and section is not considered.

4. Significant (2–3 times) change in the back tension of the last stand when the process parameters change significantly. This leads to looping or slippage of the stand rolls, disrupting the continuous rolling.

The considered system allows reducing sectional deviations caused by only one process parameter–interstand tension. Herewith, other process factors affect the workpiece section accuracy. As noted, deviations in the finished wire cross-section are associated with changes in the drawing throughout the stand, fluctuations in rolling forces due to variable initial billet dimensions, inhomogeneous physical and mechanical properties of the metal, temperature deviations, etc. [50,65,66]. The study results confirming this conclusion are given below.

*2.4. Experimental Research Results*

A long-term operation of the five-stand mill allowed for building an extensive database on the energy-power parameters of rolling various steel grades and quantifying the parameters affecting the longitudinal wire geometry variation. Figure 4 reflects the results of experimental studies of metal pressure on rolls depending on process factors [10]. They were obtained for steel grades 70, R6M5, Kh18M9Tb, and U12A–curves 1, 2, 3, and 4, respectively.

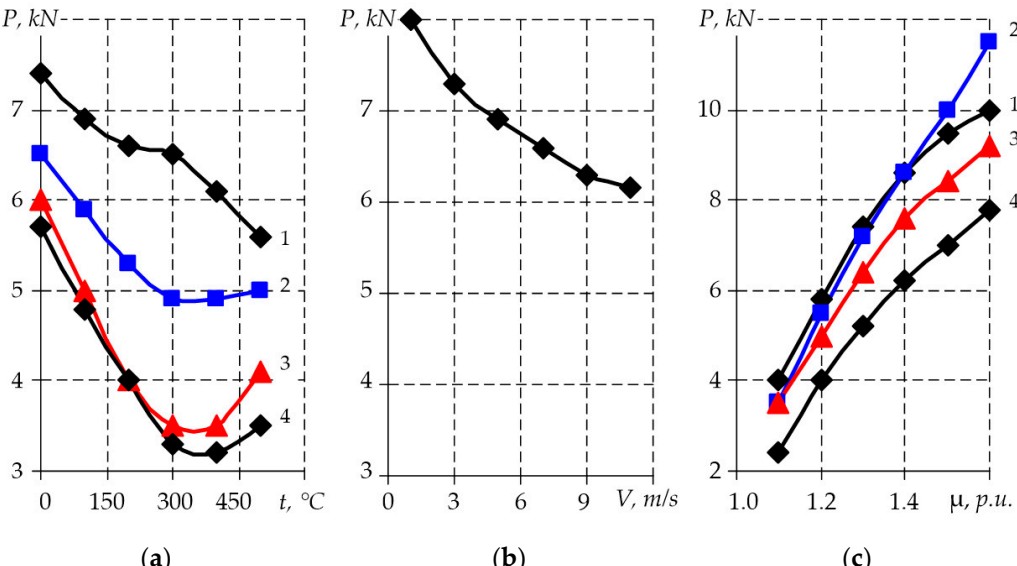

**Figure 4.** Dependencies between Metal Pressure on Roll and Semifinished Rolled Product Temperature (**a**), Rolling Speed (**b**), and Single Drawing (**c**) for Various Steel Grades.

According to Figure 4a, the pressure on the rolls reduces with an increase in the semifinished rolled product temperature. The temperature growth to 400–500 °C leads to a 20–30% decrease in pressure. The same factor explains the pressure drop with an increase in the rolling speed (Figure 4b) since the amount of emulsion contacting the wire is reduced. As a result, heat transfer reduces, and the workpiece is heated more due to the heat release caused by plastic strain. The curves in Figure 4c demonstrate that drawing affects the rolling force more than other factors listed. When this parameter changes by 30% (from 1.15 to 1.6), the pressure increases from 3 (curve 1) to ~4 (curve 4) times.

Thus, the dependencies provided confirm that process conditions non-linearly and significantly affect the metal pressure on the rolls, and, consequently, the workpiece geometry. The known automated tension control systems discussed above do not provide the section invariance to process disturbances along the workpiece length.

Along with tension control, indirect section control systems should compensate for fluctuations in the metal pressure on the rolls. Minimizing the deviation in the motor torque static component under process effects is sufficient since the rolling torque is strictly proportional to this pressure. This is a fundamentally new requirement to the system being developed. It entails an additional problem to be solved: study the pressure impact on the workpiece section.

The practical implementation and experimental study of the developed system directly at the facility are required. To confirm the technical efficiency of the implementation, the accuracy of the wire geometry control should be experimentally estimated.

The results of solving the listed problems are provided below.

## 3. Materials and Techniques
### 3.1. Impact of Pressure Deviations on Workpiece Geometry

When analyzing the longitudinal geometry variation of workpieces on continuous mills, the equations of the elastic strain of the stand and the plastic strain of the metal in the fireplace are used [67]. For a flat workpiece, these equations take the form

$$a = a_0 + \frac{P}{C_K}, \qquad (2)$$

and

$$P = f(D, a, R, \sigma_S, f, F_0, F_1), \qquad (3)$$

where $D$, $a$ is the workpiece thickness before and after the passage,

$a_0$ is the gap between the rolls before rolling,
$P$ is the metal pressure on the rolls,
$C_K$ is the stand rigidity modulus,
$R$ is the working roll radius,
$\sigma_s$ is the yield strength of the initial billet,
$f$ is the friction factor,
$F_0$, $F_1$ are the back and front tensions.

These equations are also valid to calculate the geometry variation and pressure in stands with multi-roll passes [5]. Fluctuations in the cross-section during rolling in stands with four-roll passes are determined by the change in the gap between the vertical and horizontal rolls and the pass filling degree (Figure 5). When estimating the workpiece geometry variation during rolling in such stands, its dimensions on the roll side ($a_v$, $a_h$ are the distances between vertical and horizontal roll pairs) and in the gaps between them ($d_l$, $d_r$ are the dimensions along the left and right pass diagonals) should be controlled.

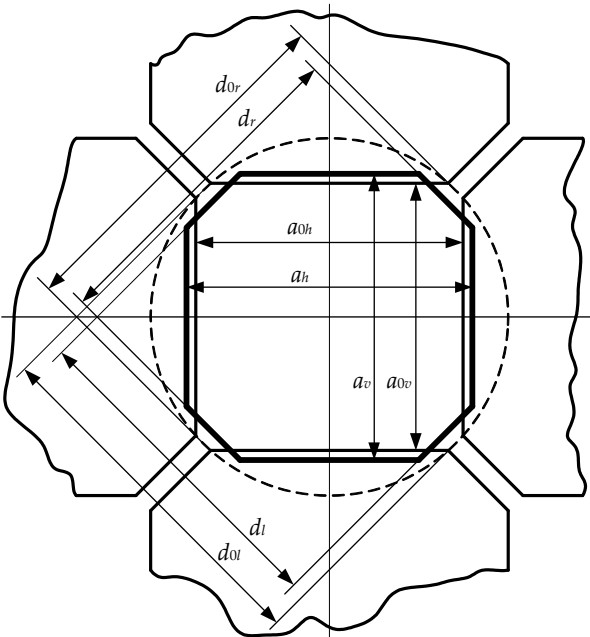

**Figure 5.** Explanations for Deriving the Longitudinal Geometry Variation Equations.

The workpiece height $a_v$ and width $a_h$ after each stand are determined by the equations

$$a_v = a_{0v} + \frac{P_v}{C_{Kv}}; \; a_h = a_{0h} + \frac{P_h}{C_{Kh}} \tag{4}$$

where $a_{0v}$, $a_{0h}$ are the distances between vertical and horizontal roll pairs along the pass symmetry axes,

$P_v$, $P_h$ are the metal pressure on vertical and horizontal rolls,
$C_{Kv}$, $C_{Kh}$ are the stand rigidity moduli in the vertical and horizontal roll planes.

To simplify the analysis, it is advisable to adopt
$C_{Kv} = C_{Kh} = C_K$; $\delta_1 = \delta_2 = \delta$; $a_v = a_h = a$; $d_l = d_r = d$.
Diagonal dimensions are determined considering the draft

$$d_l = d_{0l}(1 + \delta_1); \; d_r = d_{0r}(1 + \delta_2), \tag{5}$$

where $d_{Ol}$, $d_{Or}$ is the inscribed circle diameter for the target section along the left and right pass diagonals (Figure 5);

$\delta_1, \delta_2$ is the diagonal draft:

$$\delta_1 = \frac{d_1 - d_{0l}}{d_{0l}}; \ \delta_2 = \frac{d_r - d_{or}}{d_{or}}. \tag{6}$$

When rolling symmetrical sections from a calibrated billet, we can assume that $P_v = P_h = P; a_{0v} = a_{0h} = a_0; \ d_{0l} = d_{0r} = d_0$.

Considering the assumptions, the system of differential equations for the longitudinal geometry variation of the workpiece rolled in a four-roll pass takes the form

$$\begin{cases} da = da_0 + \frac{dP}{C_K} - \frac{P}{C_K}\frac{dC_K}{C_K}; \\ dd = dd_0 + d_0 d\delta + \delta dd_0. \end{cases} \tag{7}$$

According to Equation (7), fluctuations in the section geometry in the stands are determined by deviations (as a rule, temperature) in the metal-free pass dimensions, the pass filling degree, and the metal pressure on the rolls. The first component can be neglected since low rolling temperatures and intense roll cooling prevent a change in the distance between the rolls. During operation, it has also turned out that the workloads fall on the linear area of the stand load characteristic [3]. Therefore, the stand stiffness modulus can be assumed to be constant.

Thus, the pressure change depending on the process parameters is the key factor causing fluctuations in the product cross-section. Therefore, it can be assumed that

$$da = dP/C_K. \tag{8}$$

Considering Equation (3), the pressure change differential equation takes the form

$$dP = \frac{\partial P}{\partial D}dD + \frac{\partial P}{\partial a}da + \frac{\partial P}{\partial R}dR + \frac{\partial P}{\partial \sigma_S}d\sigma_S + \frac{\partial P}{\partial f}df + \frac{\partial P}{\partial \sigma_0}d\sigma_0 + \frac{\partial P}{\partial \sigma_1}d\sigma_1, \tag{9}$$

where $\sigma_0, \sigma_1$ are the back and front specific tensions.

In general, changes in the initial billet geometry, roll radii, and output dimensions may be characterized by a change in the drawing per pass. Therefore, Equation (9) can be reduced to the form

$$dP = \frac{\partial P}{\partial \mu}d\mu + \frac{\partial P}{\partial \sigma_S}d\sigma_S + \frac{\partial P}{\partial f}df + \frac{\partial P}{\partial \sigma_0}d\sigma_0 + \frac{\partial P}{\partial \sigma_1}d\sigma_1. \tag{10}$$

The partial derivatives, included in this equation, were calculated according to [68]. The specifics of this approach comprise the experimental determination of the equation coefficients directly at rolling. In contrast to the simulation results, this provides a high accuracy of further calculations. This approach also does not require the search for boundary conditions. As a result, formulas were obtained to determine partial derivatives when steel 70 is rolled according to the circle-square-square-circle scheme.

For the circle-square system:

$$k_0 = \partial P/\partial \mu = -1.54\mu + 35; \ k_1 = \partial P/\partial \sigma_S = -0.17\mu^2 + 0.57\mu - 0.4;$$
$$k_2 = \partial P/\partial f = 28.5\mu^2 - 42.3\mu + 13.8; \ k_3 = \partial P/\partial \sigma_0 = -0.12\mu^2 + 0.4\mu - 0.28; \tag{11}$$
$$k_4 = \partial P/\partial \sigma_1 = -0.125\mu^2 + 0.36\mu - 0.24.$$

For the square-square system:

$$k_0 = -16.2\mu + 33.4; \ k_1 = -0.08\mu^2 + 0.3\mu - 0.22;$$
$$k_2 = 19.1\mu^2 - 27.2\mu + 8.1; \ k_3 = -0.067\mu^2 + 0.27\mu - 0.205; \tag{12}$$
$$k_4 = 0.256\mu^3 - 0.02\mu^2 + 0.08\mu - 0.057.$$

For the square-circle system:

$$k_0 = -90\mu + 120; \; k_1 = -0.43\mu^2 + 1.1\mu - 0.67;$$
$$k_2 = 14.3\mu - 14.3; \; k_3 = -0.31\mu^2 + 0.81\mu - 0.504; \tag{13}$$
$$k_4 = -0.37\mu^2 + 0.89\mu - 0.52.$$

Figure 6a–c show, respectively, dependencies $k_0, k_1, k_2, k_3, k_4 = f(\mu)$ calculated by Equations (11)–(13). The vertical axis designations are shown in increments. Their analysis has shown that the impact degree of the factors being studied grows with the draft increase.

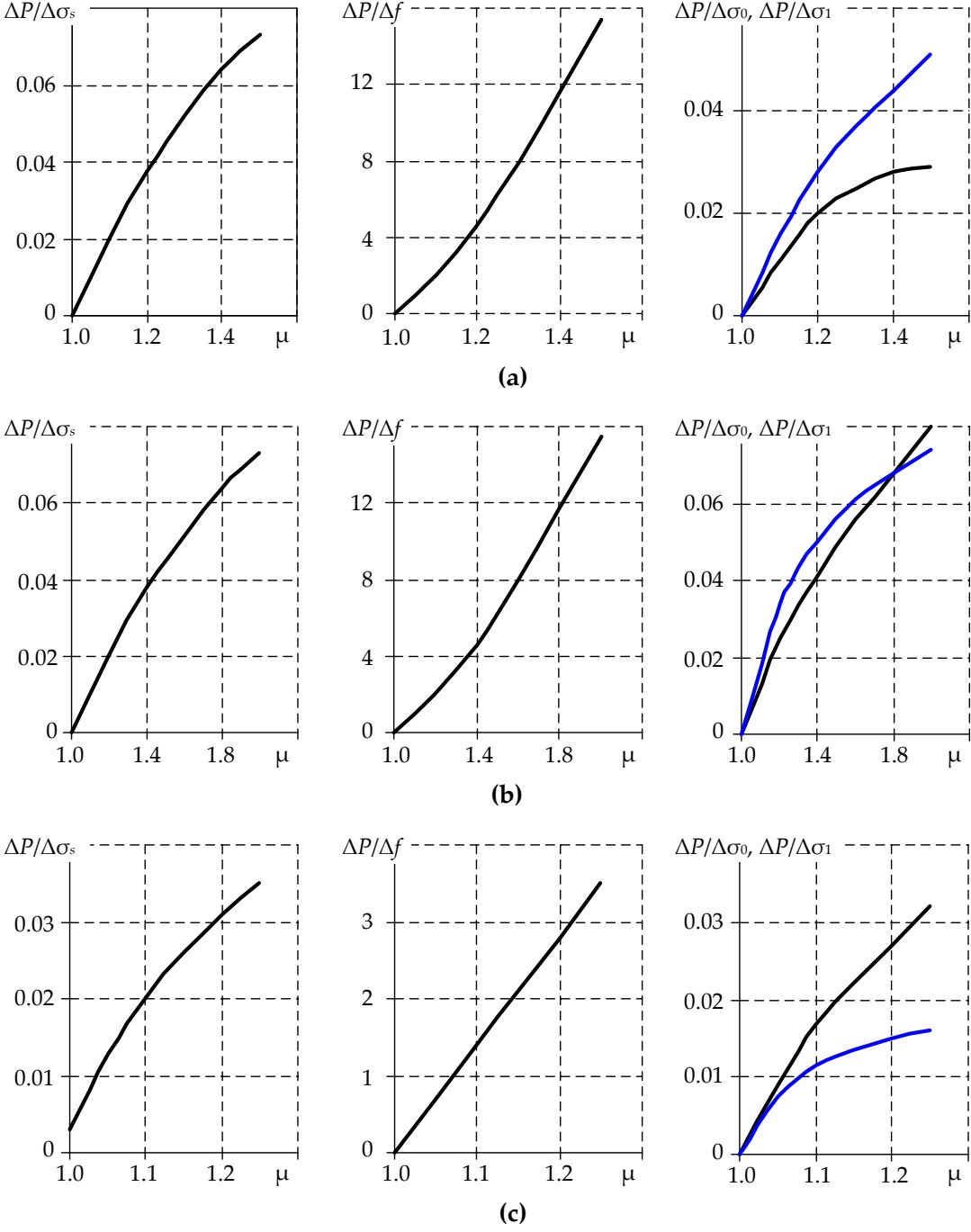

**Figure 6.** Dependencies of Partial Derivatives on Drawing When Rolling Steel 70 for the Pass Systems "Circle-Square" (**a**); "Square-Square" (**b**); "Square-Circle" (**c**).

The most significant factors causing pressure changes are the initial billet dimensions and the workpiece tension in the interstand spaces; moreover, these effects are opposite. Calculations have shown that for the unfavorable case when all factors affect the pressure along the workpiece length, the maximum pressure change is:

- 93 kN when rolling in four-roll passes according to the circle-square system,
- 116 kN when rolling according to the square-square system,
- 41 kN when rolling according to the square-circle system.

### 3.2. Continuous Rolling Condition

The stable operation of any continuous mill is conditioned by the constancy of the instant metal volumes rolled in successive stands [69]

$$Q_{M1} \cdot V_{M1} = Q_{M2} \cdot V_{M2} = \ldots Q_{M1i} \cdot V_{Mi}, \tag{14}$$

where $Q_{M1i}$, $V_{Mi}$ are the metal cross-section and speed in the $i$-th stand.

Speed mode violation affects the process course, the metal strain, and the motor load. The mill operation practice shows that when the speed changes according to the process settings, tension fluctuations of up to 50 kN are possible at standard values of 10 kN.

### 3.3. System Requirements

The foregoing allows claiming that strict contradictory requirements are set for the continuous wire rod mill drives and control system:

- on the one hand, they should assure high accuracy rolling speed control,
- on the other hand, they should maintain the set interstand tensions.

Similar to most similar units, the five-stand mill operating conditions predetermine additional requirements for the developed rolled product tension and section automated control system:

- indirect tension control in the lack of any physical tension meters,
- relative simplicity, which allows its implementation on existing mills with multi-roll passes, regardless of their design, stand number, and drive current,
- the absence of complex computational algorithms implemented in the industrial controller-based automated mill control system.

Fulfilling these conditions will allow the system development and commissioning at minimal cost, in a short time, and without additional staff training. Under the conditions of hardware and metallurgical plants, these requirements are critical.

## 4. Implementation

### 4.1. Developing a Way to Control the Rolled Product Section

The above conflicting requirements are most simply met by dividing the functions of the continuous mill process line drives. As in the existing system (Figure 3), the drive of one stand is adopted as the master, and the rest are slaves. The master drive is in charge of approaching the mill's target speed. The slave drive control systems stabilize the drive motor torques at set levels.

The fundamental difference is the compensation for deviations of process parameters by stabilizing the static component of the motor torque. The workpiece dimensions after the $i$-th stand in a plane perpendicular to the rolls are determined by Equation (2). According to Equation (8), the pressure change is the key factor causing fluctuations in the section geometry along the rolled product length. Energy and power parameters (force, torque, tension) are rigidly bound for continuous rolling. The motor torque is defined as the sum of the components

$$M_{mi} = M_{fri} + M_{Fi-1} - M_{Fi} + M_{din_i}, \tag{15}$$

where $M_{fri}$ is the free-rolling torque without considering the back and front tensions, $M_{F_i}$, $M_{F_{i-1}}$ are the torques of the front and back tension forces, reduced to the motor shaft, and $M_{din_i}$ is the dynamic torque.

The front tension torque component

$$M_{Fi} = M_{fri} + M_{Fi-1} + M_{din_i} - M_{mi}. \tag{16}$$

Respectively, tension

$$F_i = \frac{M_{F_i}}{R_i}, \tag{17}$$

where $R_i$ is the radius of rolls of the *i*-th stand.

It follows that the constant $M_{F_i}$ torque's static component determines the stable tension in all modes.

The proposed method is based on maintaining the slave motor static torques constant regardless of the process conditions. When changing the rolling parameters, the interstand tension constancy effect is achieved by controlling torques. Threat, tension changes contribute to reducing fluctuations of metal pressure on the rolls. This will reduce the workpiece's longitudinal geometry variation caused by the process factor deviations. As a result, the accuracy of shaping a target section along the length of the finished rolled product will be improved.

Due to these features, the developed technique was called the Rolled Product Section Automated Control System. This name was assigned by analogy with the (hot- or cold-rolled) flat product section control. The sheet mill RPSACS is aimed at providing a rectangular strip cross-section at the last stand output [70,71]. The RPSACS may comprise systems for automated control of tension, thickness (as a rule, based on hydraulic roll screw-down), flatness, etc. [72–75] Mills with multi-roll passes are not equipped with such systems; therefore, RPSACS should be based on drive speed and torque automated control systems. Under the constant risk of pass stalling, complex dynamic algorithms virtually cannot be implemented on the basis of such systems. This justifies the requirement for the developed system simplicity.

### 4.2. Five-Stand Mill Control System

Figure 7 shows the functional circuit of the drive control system implementing this method, developed for the mill being studied. It differs from the system shown in Figure 3 by selecting the second stand drive as the master one. This solution has been justified in the course of the system configuration. The experimental study has shown that the first or last stand drive should not be the master one since in this case one of the tension components (back or front) is missing. This violates the system settings at the gauge changing.

The stand drives are made according to the FC-AM system with vector coordinate control. As noted above, the reconstruction supposes installing such drives. The master stand drive has a speed control system ASCS and the rest–automated torque control systems ATCS. Ref. [63] describes the transfer function calculation technique for the controllers of these systems. The signal designations correspond to those given in the Figure 3 legend.

At the wire rod loading stage, the speed control drives are controlled manually. In this mode, the speed setting signals are corrected, which eliminates both loopings in the interstand spaces and high tensions. Manual control is retained even after loading when the mill accelerates to intermediate speed. Upon reaching a stable mode, a command is given to fix the slave-stand motors torques. When the system switches to automated control, they become signals setting the drive torque. Then, the torque controllers are put into operation, the mill is synchronously accelerated to the operating speed, and further rolling is performed while automatically maintaining the set torques. Control signals are generated in the speed and torque setting module, implemented in the mill's APCS based on industrial controllers (PC). There are no special requirements for the PC resource.

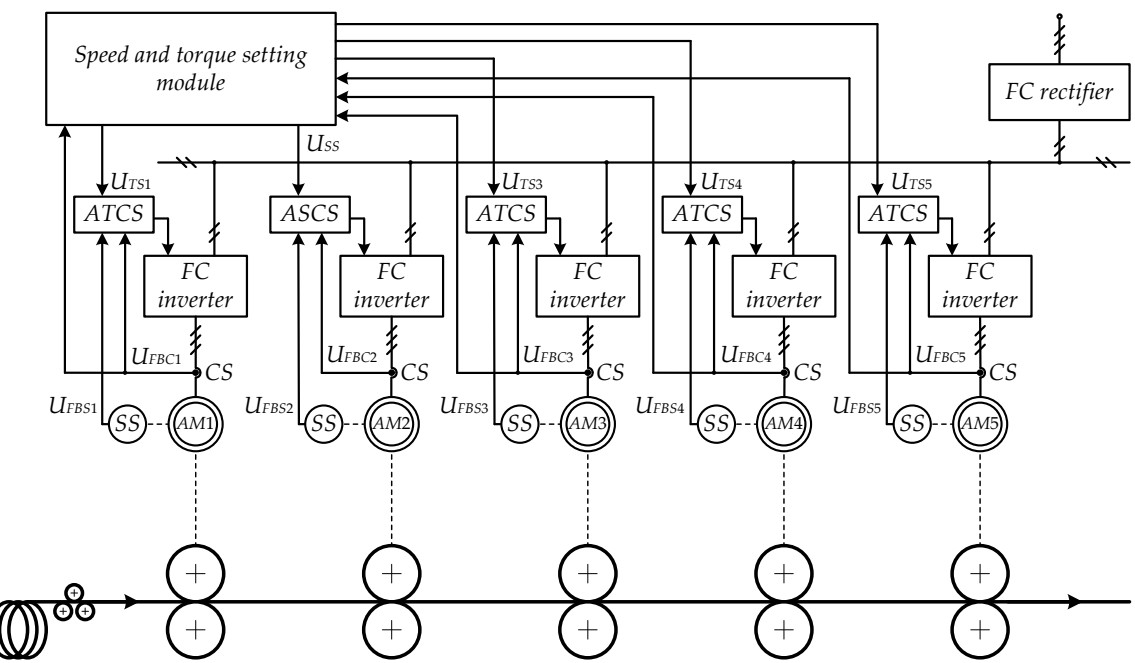

**Figure 7.** Functional Circuit of a Five-Stand Mill RPSACS: speed and torque setting module; frequency converter (FC) rectifier; automated torque control system (ATCS); automatic speed control system (ASCS); FC inverter; $U_{FBi}$, $U_{FBi}$-current and speed feedback voltages; CS–current sensor; AM-asynchronous motor; SS-speed sensor; $U_{TSi}$-torque setting voltages.

## 5. Results

### 5.1. Experimental Research

The algorithm for the developed control technique was implemented in the five-stand mill PC software. A series of experiments were performed to estimate the technical efficiency of implementing the proposed technique to control sections. The experiments were performed with thyristor DC drives installed on the mill. However, from the standpoint of optimizing control algorithms external for a closed-loop drive coordinate control system, the current type is immaterial. The processes occurring in the closed-loop speed or torque control systems are identical for both VFDs with vector control and DC drives [69,76–78]. Therefore, it can be argued that implementing the FC-AM system will produce similar results.

Figure 8a,b show, respectively, oscillograms obtained for the existing and developed control algorithms in the mill acceleration mode with metal in the rolls. The motor armature current $i_1$ was fixed, which was proportional to the motor torque when operating without field weakening. Along with the $i_1$ current oscillograms, those for the $T_1$ and $T_2$ tensions in the first and second interstand spaces and the first stand drive speed $\omega_1$ are given.

The frequent case of acceleration with insufficiently accurate setting of the successive stand drive initial speeds was studied. The 10% speed mismatch has been fixed, without virtually affecting the loading speed. However, for the existing system (Figure 8a), interstand tensions change significantly as the mill accelerates. In the first interstand space, the tension decreases by $\Delta T_1 = 4.65$ kN, which is about 60% of the initial value. In the second gap, the deviation $\Delta T_2 = 4.2$ kN is 84% of the set value. This confirms that in the worst case, the tension may approach zero, which poses the risk of looping.

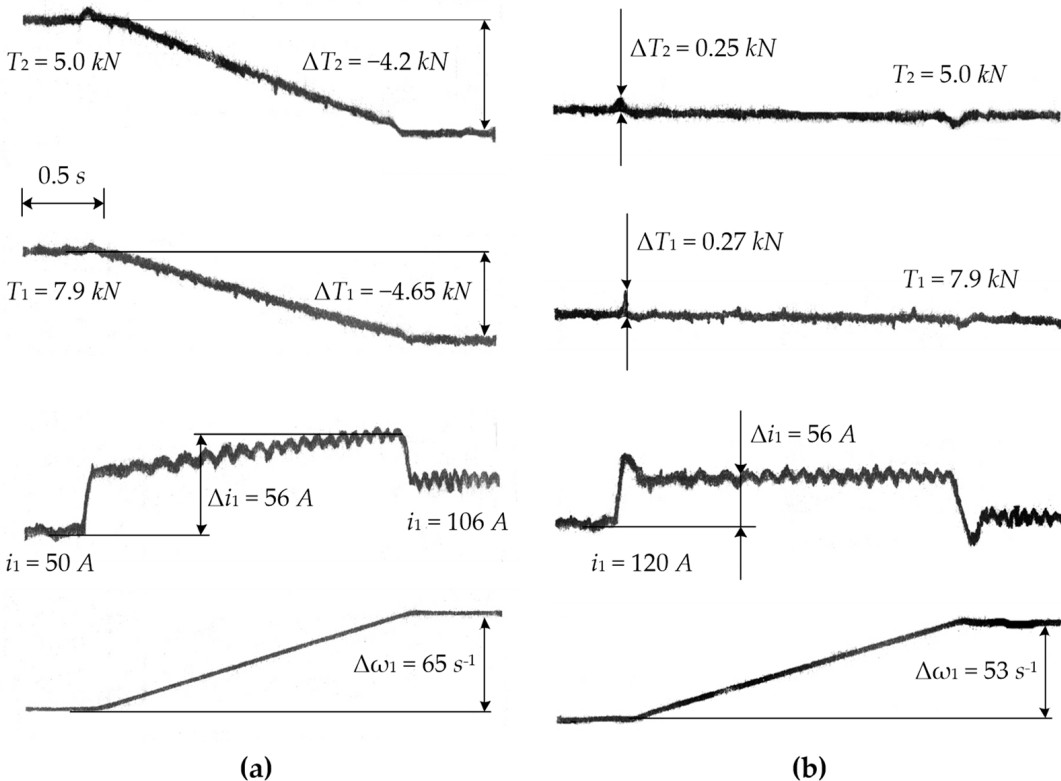

**Figure 8.** Mill Acceleration Transient Processes for the Existing (**a**) and Developed (**b**) Control Systems.

Oscillograms for the RPSACS proposed (Figure 8b) were obtained with similar initial speed ratios. According to them, in the mill acceleration mode, the interstand space tensions $T_1$ and $T_2$ remain constant. The dynamic motor current $i_1$ does not change significantly. This is an indirect sign of no tension impact on the pressure and, accordingly, the workpiece geometry at the first stand output. The drawback is decreasing the steady drive speed caused by the torque control. However, this drawback is eliminated by additional correction of the master drive speed.

Figure 9a shows transient process oscillograms for the same coordinates, obtained in a quasi-steady rolling mode with the first stand current setting changed. The second stand speed and current settings remained the same. This mode simulates the occurrence of the workpiece geometry variation (increase in the diameter) at the first stand input. The tension mismatch is set to 2.6 kN ($T_1$ = 8.05 kN, $T_2$ = 5.45 kN), whereas the current $i_1$ = 41 A. After responding to the disturbance, a new current value of 119 A is set. The first stand front (back for the second one) tension $T_1$ decreases by $\Delta T_1$ = 5.9 kN to 2.15 kN. To maintain the target current at the second stand, the ATCS affects the first stand drive speed $\omega_1$ towards increasing. As a result, the interstand space tension decreases by $\Delta T_2$ = 2.6 kN to 2.85 kN. The tension difference decreases by 3.5 times to 0.7 kN. Thus, the existing ATCS controls tension indirectly with satisfactory accuracy.

Oscillograms in Figure 9b characterize the proposed system operation when simulating an increase and further decrease in the wire rod diameter at the stand input. These disturbances are tested with a fast response, and the duration of current transients does not exceed 0.5 s. The steady-state tensions are reached for the same time without overshoots and fluctuations. The system changes the interstand tensions towards reducing the pressure deviations caused by the disturbance (the figure does not provide pressure oscillograms). This is an indirect confirmation of reduced tension impact on the rolled product geometry when using the RPSACS proposed. The results of direct measurements confirming this conclusion are provided below.

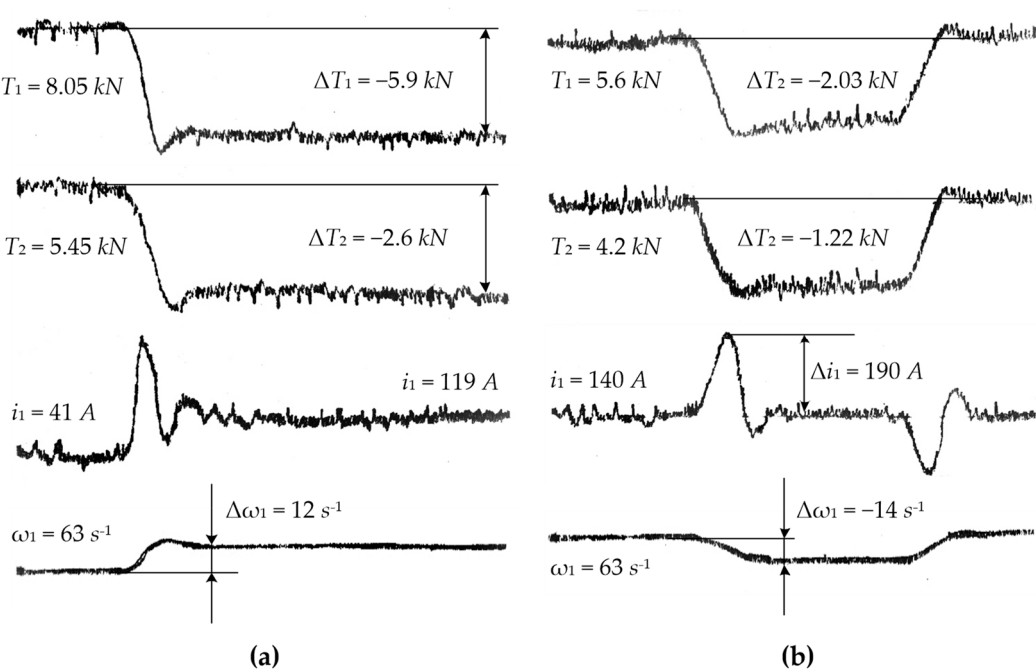

$T_1 = 8.05\ kN$          $\Delta T_1 = -5.9\ kN$

$T_2 = 5.45\ kN$          $\Delta T_2 = -2.6\ kN$

$i_1 = 41\ A$                $i_1 = 119\ A$

$\omega_1 = 63\ s^{-1}$        $\Delta\omega_1 = 12\ s^{-1}$

$T_1 = 5.6\ kN$          $\Delta T_2 = -2.03\ kN$

$T_2 = 4.2\ kN$          $\Delta T_2 = -1.22\ kN$

$i_1 = 140\ A$          $\Delta i_1 = 190\ A$

$\omega_1 = 63\ s^{-1}$          $\Delta\omega_1 = -14\ s^{-1}$

**(a)**          **(b)**

**Figure 9.** Transient Processes When Simulating a Change in The Semifinished Rolled Product Diameter in The Existing (**a**) & Developed (**b**) Control Systems.

### 5.2. Thickness Equalization Factor

The impact of changes in the initial wire rod diameter on fluctuations in the rolling section geometry is generally estimated by the equalization factor [5]. It is defined as the ratio of relative fluctuations in the section geometry before and after rolling. For the case of symmetrical rolling in four-roll passes, the equalization factor is defined by the following equation

$$k_E = \frac{\Delta d_{cs}/d_{cs}}{\Delta d_{is}/d_{is}}, \tag{18}$$

where $d_{cs}$ is the diameter of the circle circumscribing the semifinished rolled product section at the stand input,

$d_{is}$ is the diameter of the circle inscribed in the workpiece section after rolling, and $\Delta d_{cs}$, $\Delta d_{is}$ are the changes in these diameters.

In the general case, the equalization factor can take the following values:

- $k_E > 1$—in this case, the initial relative deviation in the section geometry is greater than the final one, and the stand reduces the section geometry fluctuations along the workpiece length,
- $k_E = 1$—the initial and final relative dimensions are equal, and rolling in the stand does not affect the workpiece section deviations,
- $k_E < 1$—the initial relative deviation in the section geometry is less than the final one, and the stand enhances the section geometry fluctuations.

To assess the longitudinal geometry variation, an experiment was performed, which involved the rolling of three wire grades differing in the stiffness modulus. Figures 10–12 show the results of measuring the wire rod diameters at the mill input and the round section obtained after the rolling. The section dimensions are conditionally shown in Figure 10a. Diagrams in Figure 10b show that initially, the wire rod had an ovality of 0.3 mm, and the maximum section deviation was 0.4 mm. After rolling, the maximum round section deviation does not exceed 0.05 mm. It was found that the geometry variation was mainly caused by whipping supports and working roll disks.

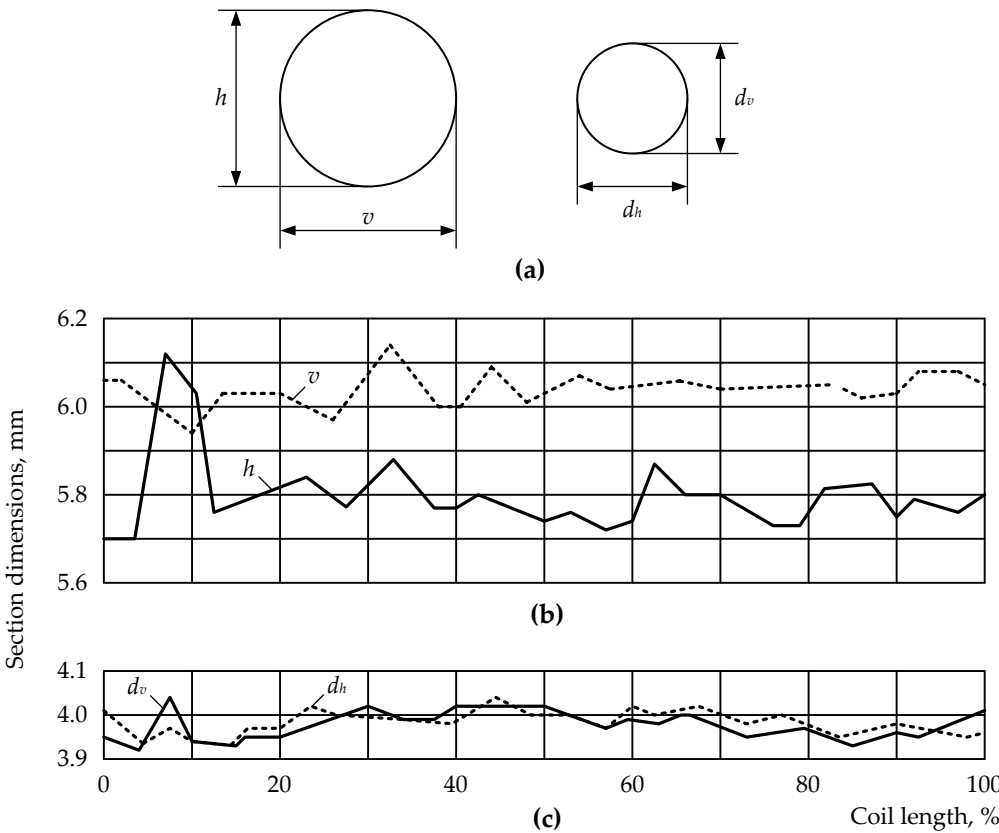

**Figure 10.** The Circumscribed and Inscribed Circles Explaining the Changes in the Workpiece Diameter (**a**); The Longitudinal Geometry Variation Measurement Results for rhe Initial Billet (**b**) and Wire (**c**). Steel R6M5: Section Dimensions, mm; Coil Length.

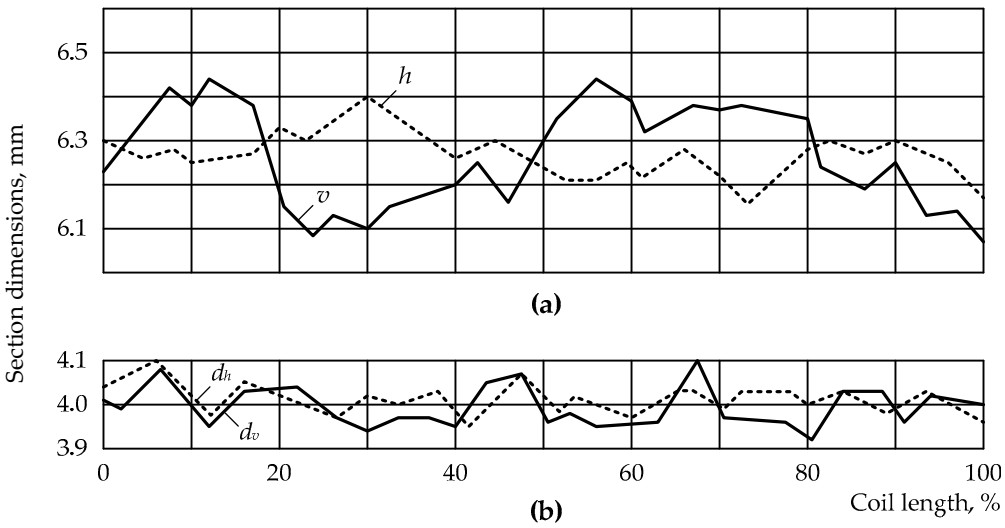

**Figure 11.** Diagrams of Longitudinal Geometry Variation for the Initial Billet (**a**) and Wire (**b**). Steel U12A.

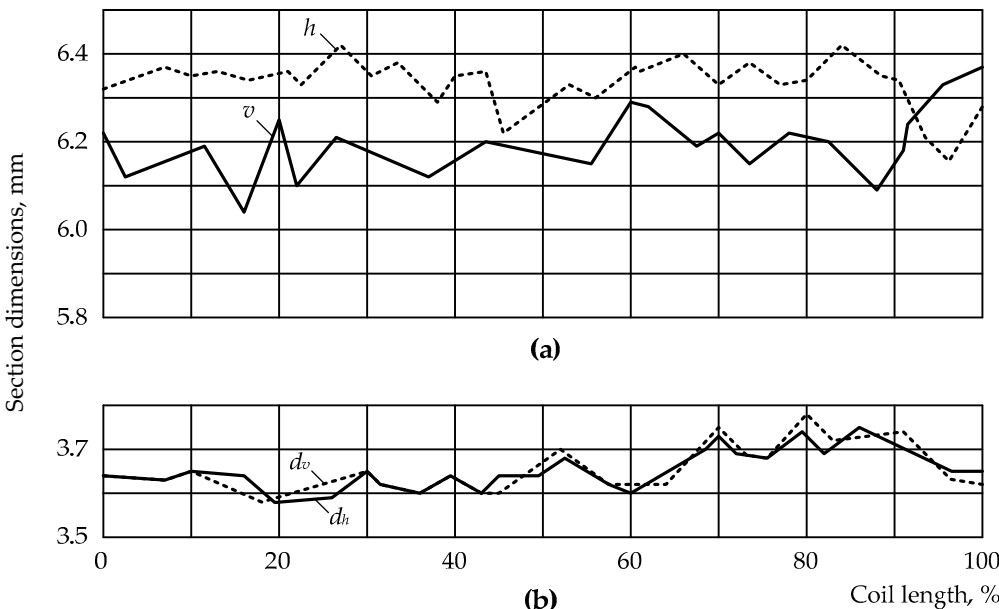

**Figure 12.** Diagrams of Longitudinal Geometry Variation for the Initial Billet (**a**) and Wire (**b**). Steel KhI8N9T.

The results of the analysis for the diagrams in Figures 11 and 12 allow for drawing similar conclusions.

## 6. Discussion of the Results

To generalize the experimental research results, averaged equalization factors were calculated for the rolling of 20 coils with the 3.2 mm diameter wire from a 6 mm diameter wire rod made of steel grades 70 and 80. The pass and drawing shapes by stands corresponded to those specified in Table 1; the total drawing in five stands $\mu_\Sigma$ = 3.5. The wire rod and workpiece diameters were measured, respectively, at the input and output of each stand using a D20 portable laser thickness gauge available at the enterprise. The device allows measuring the current diameter and the value of exceeding the set range with reference to the length. The device measuring heads were installed alternately in the interstand spaces. The measured thickness range was 0.1–20 mm; 120 measurements per second were made; the error was ±1.5%. Averaged measurement results are provided in Table 2. It also provides the results of calculating relative draft and equalization factors after each stand.

**Table 2.** Averaged Geometry Variation at Batch Rolling.

| Parameters | Semifinished Rolled Product Dimension | Circle-Square | Circle-Octagon | Octagon-Square | Octagon-Octagon | Octagon-Circle |
|---|---|---|---|---|---|---|
| Drawing ratio | - | 1.57 | 1.11 | 1.66 | 1.17 | 1.05 |
| Inscribed circle diameter, mm | 6.0 | 4.8 | 4.6 | 3.55 | 3.3 | 3.2 |
| Maximum section diameter, mm | 7.56 | 5.69 | 5.28 | 3.98 | 3.65 | 3.5 |
| Absolute draft, mm | - | 1.2 | 0.2 | 1.05 | 0.25 | 0.1 |
| Relative draft, % | - | 20 | 4 | 23 | 7 | 3 |
| Final size deviation, mm | 1.56 | 0.89 | 0.68 | 0.43 | 0.35 | 0.3 |
| Equalization factor | - | 1.4 | 1.29 | 1.2 | 1.15 | 1.1 |

The results of the analysis for the diagrams in Figures 10–12 and the data in Table 2 allow for drawing the following conclusions:

1. In all cases, the equalization factor takes values greater than one. The equalization factor increases (not reflected in the calculations) with an increase in the stiffness modulus and changes slightly (within 1.1–1.4, see the last table row) with the draft decrease.

2. A favorable draft scheme at the rolling in four-roll passes allowed for obtaining a product with the cross-section fluctuations not exceeding 0.43 mm already after the third stand. Thus, the initial deviation (1.56 mm) is reduced by more than 3.5 times. Averaged finished wire geometry variation is reduced by more than 5 times.

In general, the research allowed for stating that when implementing the developed rolled product tension and section control technique, disturbances such as initial billet section variability, friction factor changes, and strain resistance lead to a reverse change in the front tension. This compensates for the impact of the listed factors on the metal pressure on the rolls and the rolling torque. As a result, deviations of the finished workpiece section from the target one are reduced.

## 7. Conclusions

Note that rolling on continuous mills with multi-roll passes is a promising steel wire production technology. The key condition for maintaining the wire geometry variation and ovality within tolerances is a high accuracy of shaping workpiece sections at each stand output.

It has been established that the factors most affecting the section geometry in stands with multi-roll passes are interstand space tensions and metal pressure on the rolls. In the practice of operating the continuous bar and wire rod mills, fluctuations in the semifinished rolled product geometry in the continuous stand groups are eliminated by tension adjustment. Known ATCS based on the free-rolling current or zero tension control do not compensate for the impact of pressure changes on the workpiece section.

A five-stand mill with four-roll passes, operating at the Beloretsk Metallurgical Plant, has been characterized. Drawbacks of the existing ATCS have been identified, the major of which are:

- the control algorithm complexity and insufficient section geometry control accuracy,
- no compensation for the impact of disturbances on the metal pressure on the rolls, and no limitation of the metal pressure impact on the workpiece geometry,
- the system allows for reducing section deviations caused by only one process factor—interstand tension.

The results of the experimental research on metal pressure are provided for rolling in four-roll passes. Pressure dependencies on semifinished rolled product temperature, rolling speed, and drawing in the stand have been studied for various steel grades. It has been found that drawing in the stand has the greatest impact on the rolling force. When this parameter changes by 30%, the pressure increases by 3–4 times, depending on the steel grade.

The requirements for the developed RPSACS have been formulated. The system should control tension and limit the impact of external factors on the metal pressure on the rolls. It is shown that to fulfill the second requirement, the motor torque static component deviations should be minimized. The rolled product tension and section automatic control system is considered, which has been developed for the five-stand mill being studied. Its fundamental difference from the existing ATCS configured for zero current control is the compensation for deviations in process parameters by stabilizing the motor torque static components. This is achieved by dividing the stand drives into the master and slave ones designed for, respectively, closed-loop speed and motor torque control. When implementing the developed system, the mill control algorithm provides fulfilling the requirements for RPSACS.

A series of experiments were performed to estimate the technical efficiency of implementing the developed RPSACS. A comparative analysis of oscillograms is provided for the existing and developed systems. In the second mode (compensation for geometry variation), the disturbance was tested with a fast response. The tensions reach steady val-

ues without overshoots and fluctuations. The system adjusts interstand tensions towards reducing the workpiece geometry variation caused by changes in process conditions.

The results of direct measurement conducted for the semifinished rolled product and finished wire cross-section deviation in the vertical and horizontal direction are provided for three steel grades. As a criterion, the stabilization factor is taken, defined as the ratio of relative fluctuations in the diameters of the circles inscribed in the section, calculated before and after rolling. The authors concluded that in all cases, this factor takes values greater than one, i.e., the geometry variation reduces. The optimal draft scheme also allows for obtaining a satisfactory workpiece section already at the third stand output of a five-stand mill.

In general, theoretical and experimental studies allow for stating that implementing a technology for new rolled product tension and section automated control reduces the impact of disturbances on the rolling torque and metal pressure on the rolls. As a result, the wire section deviations are reduced, which confirms the achievement of the goal set.

The development is recommended for the application on the wire, bar, and sheet rolling mills.

**Author Contributions:** Methodology, A.S.K. and A.A.R.; ideas A.A.R. and O.I.P.; software I.N.E.; validation V.R.K.; formal analysis V.R.K. and B.M.L. All authors have read and agreed to the published version of the manuscript.

**Funding:** Research was funded by the Ministry of Science and Higher Education of the Russian Federation under a subsidy for a government-funded basic research project, Contract No. FENU-2020-0020 (2020071GZ).

**Conflicts of Interest:** The authors declare no conflict of interest.

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
