# Peer review of "Developing an Automated System to Control the Rolled Product Section for a Wire Rod Mill with Multi-Roll Passes"

_jmmp, doi:10.3390/jmmp6040088_

Round 1
Reviewer 1 Report
1. This paper has proposed a control system development that is capable of obtaining more stable wire rolling process and smaller product geometry variation. The experimental results of the tension and the current variations in stand 1 and 2 had shown the better stability compared with the existing ATCS system.
2. The parameters “drawing” and “drafting” in Table 1 and Table 2 should be defined more clearly and give a schematic diagram of the calculation.
3. The term of “thickness variation” used in the paper should be corrected to the “geometry variation” or “maximum deviations of cross-section” since the semifinished rolled product is a wire instead of a plate.
4. The symbol “m” represents dimension unit in Figure 4 ( L302) and a variable in equation 9 (L383), it is confusing and should be modified to avoid misunderstanding.
5. The symbol Ri in equation 16 (L448) should be explained no matter the symbol R has been defined in previous section (L340).
6. The unit of tension in Figures 8 and 9 should be kN instead of H. The tension variation of stand 1 in Figure 9b DT2 = -2030 H is not correct, it should be DT1 = -2030 kN.
7. L 672-675, the initial speed mismatch had not been discussed in the previous sections and should not be included in the conclusion section.
8. L681-L684 , the longitudinal thickness variation should be modified to cross-section deviation in the vertical and horizontal direction or longitudinal dimension variation in short. The stabilization factor should be correct to equalization factor for the consistency in the paper.
9. L645, “the experimental research of metal pressure are provided”, it is not correct, in this paper, only pressure change differential equation had given, no exact data of the metal pressure were given in any table. The authors should make calculation of the metal pressure and give a table to show the pressure data explicitly.

Author Response
The authors thank the Reviewer for the close reading of the paper and comments provided.
- This paper has proposed a control system development that is capable of obtaining more stable wire rolling process and smaller product geometry variation. The experimental results of the tension and the current variations in stand 1 and 2 had shown the better stability compared with the existing ATCS system.
The authors fully agree with this conclusion.
- The parameters “drawing” and “drafting” in Table 1 and Table 2 should be defined more clearly and give a schematic diagram of the calculation.
In both tables, the ‘drawing’ parameter has been replaced by the more accurate ‘drawing ratio’. To clarify its calculation, a fragment has been added to Subsection 2.1:
According to the definition, drawing ratio (drawing) means the ratio of the workpiece lengths or cross-sections before and after rolling in the stand:
, |
(1) |
where l0, Q0 are the workpiece length and cross-section before rolling (at the stand inlet);
l1, Q1 are the workpiece length and cross-section after rolling (at the stand outlet).
Workpiece herein means an intermediate product between the primary billet (semi-finished rolling product) and the finished wire.
Hereinafter, the numbering of formulas has been changed appropriately.
- The term of “thickness variation” used in the paper should be corrected to the “geometry variation” or “maximum deviations of cross-section” since the semifinished rolled product is a wire instead of a plate.
We agree with the comment; in the paper, the term ‘thickness variation’ has been replaced by ‘geometry variation’; corrections are highlighted in color.
- The symbol “μ” represents dimension unit in Figure 4 ( L302) and a variable in equation 9 (L383), it is confusing and should be modified to avoid misunderstanding.
The ‘μ’ symbol is not a unit of measure but the designation of the drawing ratio in equation (1). The coefficient is dimensionless, i.e., measured in relative units (p.u.). As a reply to the comment, Fig. 4 has been corrected: μ, p.u.
- The symbol Ri in equation 16 (L448) should be explained no matter the symbol R has been defined in previous section (L340).
The authors agree with the comment. An explanation of formula (17) has been added to the paper and highlighted in color:
where Ri is the radius of rolls of the i-th stand.
- The unit of tension in Figures 8 and 9 should be kN instead of H. The tension variation of stand 1 in Figure 9b DT2 = -2030 H is not correct, it should be DT1 = -2030 kN.
Figs. 8 & 9 have been corrected. The text has been corrected appropriately.
- L 672-675, the initial speed mismatch had not been discussed in the previous sections and should not be included in the conclusion section.
We agree with the comment. Fragment of two sentences has been removed from the text.
It is shown that in the acceleration mode, in the existing system, when the initial speeds of the successive stand drives mismatch by 10 %, there is a risk of tension loss. In the developed system, a similar mismatch has been tested with minimum dynamic tension deviations in adjacent interstand spaces.
- L681-L684 , the longitudinal thickness variation should be modified to cross-section deviation in the vertical and horizontal direction or longitudinal dimension variation in short. The stabilization factor should be correct to equalization factor for the consistency in the paper.
The authors agree with the comment.
‘Longitudinal thickness variation’ has been replaced by 'cross-section deviation in the vertical and horizontal direction’ and highlighted in color.
- L645, “the experimental research of metal pressure are provided”, it is not correct, in this paper, only pressure change differential equation had given, no exact data of the metal pressure were given in any table. The authors should make calculation of the metal pressure and give a table to show the pressure data explicitly.
The authors do not agree with the comment. The conclusion is made based on the experimental study results represented in Fig. 4 as dependencies of the metal pressure on the rolls on the semifinished rolled product temperature, rolling speed, and single drawing for various steel grades. We consider it inappropriate to represent these graphical dependencies as a table. Any theoretical metal pressure calculation is not required since we are talking about experimental studies.
The experimental study results are also represented by oscillograms in Figs. 8 & 9. They specify the exact metal pressure. Also, the numerical pressure values are given in paragraph 5.1. Therefore, we consider the conclusion in line L645 legitimate.

Reviewer 2 Report
1. The abstract and conclusions of this article are too length, which might cause that readers cannot visually obtain the research significances of this article. It is recommed to properly shorten the length of abstract and conlucions.
2. In Table.1, the values of drawing were written as 1.57,1.11,1.66,1.17,1.05, how the word "drawing" is defined? The drawing should be defined in the article.
3. For Fig. 9, the explanations of (a) and (b) are missed, which should be added in caption.
Author Response
The authors thank the Reviewer for the close reading of the paper and comments provided.
- The abstract and conclusions of this article are too length, which might cause that readers cannot visually obtain the research significances of this article. It is recommended to properly shorten the length of abstract and conclusions.
The authors agree with the comment. The length of the abstract and conclusions has been properly shortened.
- In Table.1, the values of drawing were written as 1.57,1.11,1.66,1.17,1.05, how the word "drawing" is defined? The drawing should be defined in the article.
To clarify the ‘drawing’ term, a fragment has been added to the paper:
According to the definition, drawing ratio (drawing) means the ratio of the workpiece lengths or cross-sections before and after rolling in the stand:
, |
(1) |
where l0, Q0 are the workpiece length and cross-section before rolling (at the stand inlet);
l1, Q1 are the workpiece length and cross-section after rolling (at the stand outlet).
Workpiece herein means an intermediate product between the primary billet (semi-finished rolling product) and the finished wire.
Hereinafter, the numbering of formulas has been changed appropriately.
- For Fig. 9, the explanations of (a) and (b) are missed, which should be added in caption.
The authors agree with the comment. The caption for Fig. 9 has been replaced as follows:
Fig. 9. Transient Processes When Simulating a Change in The Semifinished Rolled Product Diameter in The Existing (a) & Developed (b) Control Systems

Reviewer 3 Report
A nice manuscript to read, with some interesting results provided. Nice balance of theory and application.
Some review of the English would be beneficial
Author Response
The authors thank the Reviewer for the high assessment of the paper.